# Targeting Protein Tyrosine Phosphatases to Improve Cancer Immunotherapies

**DOI:** 10.3390/cells13030231

**Published:** 2024-01-25

**Authors:** Robert J. Salmond

**Affiliations:** Leeds Institute of Medical Research at St. James’s, School of Medicine, University of Leeds, Leeds LS9 7TF, UK; r.j.salmond@leeds.ac.uk

**Keywords:** protein tyrosine phosphatase (non-receptor type), cancer immunotherapy, tumour suppressor, oncogene, T cells, cell signalling

## Abstract

Advances in immunotherapy have brought significant therapeutic benefits to many cancer patients. Nonetheless, many cancer types are refractory to current immunotherapeutic approaches, meaning that further targets are required to increase the number of patients who benefit from these technologies. Protein tyrosine phosphatases (PTPs) have long been recognised to play a vital role in the regulation of cancer cell biology and the immune response. In this review, we summarize the evidence for both the pro-tumorigenic and tumour-suppressor function of non-receptor PTPs in cancer cells and discuss recent data showing that several of these enzymes act as intracellular immune checkpoints that suppress effective tumour immunity. We highlight new data showing that the deletion of inhibitory PTPs is a rational approach to improve the outcomes of adoptive T cell-based cancer immunotherapies and describe recent progress in the development of PTP inhibitors as anti-cancer drugs.

## 1. Introduction

In recent decades, the development of therapies that bypass immunosuppressive pathways to enhance immune responses has resulted in improved outcomes in the treatment of cancer. Function-blocking antibodies that target immune checkpoint receptors, such as programmed death 1 (PD-1) or cytotoxic T lymphocyte antigen 4 (CTLA-4), have become the standard of care for the treatment of melanoma [1] and carcinomas of the head and neck [2], lung [3], kidney [4] and bladder [5]. Furthermore, adoptive cell therapies (ACT) using chimeric antigen receptor T cells (CAR-T) have revolutionised the treatment of haematological malignancies [6]. Despite these advances, many patients do not benefit from current immunotherapy modalities; even in “immunotherapy-sensitive” cancers, such as melanoma, ~50% of patients fail to respond to immune checkpoint inhibitors, whilst CAR-T therapy has yet to be translated successfully to the treatment of solid tumours. Therefore, the identification of novel targets to improve immune responses in cancer stands to bring benefit to the majority of cancer patients.

Protein tyrosine phosphatases (PTPs) are a diverse family of enzymes that play fundamental roles in the propagation, regulation and termination of intracellular signalling pathways. For a general overview of the biological functions of the PTPs, we recommend excellent review articles [7,8,9,10]. At the most basic level, PTPs act to counteract the activity of protein tyrosine kinases (PTKs) by removing phosphate groups from phosphorylated tyrosine residues in signalling proteins. PTPs may have functions independent of their phosphatase activity and typically have additional functional domains that regulate their intracellular localisation and interaction with binding partners. Important roles for PTPs in the regulation of cancer cell biology and immune responses to tumours have emerged. In this review, we describe the role of PTPs, in particular non-receptor type (PTPN) family members, in these processes with a focus on recent advances in targeting PTPs as an approach to improving cancer immunotherapy.

## 2. PTPN1 and PTPN2

### 2.1. Roles of PTPN1 and PTPN2 in Cell Signalling

PTPN1, commonly referred to as PTP1B, and PTPN2, referred to as T cell PTP (TC-PTP) in some of the literature, are related phosphatases with a high degree of sequence homology, particularly in their catalytic domains [11]. The basic domain structure of PTPN1, PTPN2 and the other PTPs discussed in the current work is given in Figure 1. PTPN1 is expressed in a variety of tissues and plays an important role in metabolic signalling. In this regard, mice lacking PTPN1 display enhanced insulin sensitivity and resistance to weight gain when fed a high-fat diet [12]. PTPN2 is also ubiquitously expressed and plays a vital role in the regulation of Janus kinase (JAK)—signal transducer and activator of transcription (STAT) signalling downstream of a wide variety of cytokine and growth factor receptors. Germline deletion of *Ptpn2* in mice results in stunted growth, haematopoietic defects, anaemia, systemic inflammation and the death of homozygous animals by 5 weeks postnatally [13,14]. In T cells, PTPN2 is a negative regulator of both T cell receptor (TCR) signalling and JAK-STAT pathways [15,16], whilst *PTPN2* polymorphisms have been identified as risk factors for the development of autoimmunity [17].

### 2.2. PTPN1 and PTPN2 as Regulators of Cancer Cell Biology

Evidence for cancer cell-intrinsic roles for PTPN1 and PTPN2 in the development or suppression of tumour growth has emerged. For example, PTPN1 plays a positive role in human epidermal growth factor receptor 2 (HER2) signalling during breast cancer development [18]. Pharmacological inhibition of PTPN1 using a small molecule inhibitor MSI-1436 antagonises HER2 signalling and blocks the growth of breast cancer xenografts in mice [19]. In contrast, PTPN1 can also act as a tumour suppressor. Thus, lineage-specific deletion of *Ptpn1* using LysM-Cre results in the development of myeloid leukaemias [20]. Development of leukaemia in myeloid-specific *Ptpn1*-deficient mice was associated with hyperphosphorylated STAT3 and JAK-dependent signalling. Similarly, PTPN2 acts as a tumour-suppressor protein in T cell leukaemias and triple-negative breast cancer (TNBC) by counteracting oncogenic Src kinase and JAK-STAT signalling [21,22,23]. Furthermore, loss of PTPN1 and PTPN2 expression was recently identified as an important driver of anaplastic large cell lymphoma (ALCL) resistance to anaplastic lymphoma kinase (ALK) inhibitors [24]. Thus, both PTPN1 and PTPN2 regulate ALK phosphorylation and activity, whilst ALCL patients who had developed resistance to ALK inhibitors demonstrated downregulation of PTPN1/PTPN2 expression. PTPN1 also acts to dephosphorylate another tyrosine phosphatase family member, PTPN11 [24], which is a positive regulator of ALK-dependent RAS/mitogen-activated protein kinase (MAPK) signalling [25]. Consequently, the loss of PTPN1 and/or PTPN2 expression in ALCL results in hyperactive PTPN11, MAPK and JAK-STAT signalling [24].

Evidence suggests that PTPN2 plays a key role in regulating the cell-intrinsic sensitivity of tumour cells to immunotherapy. An in vivo CRISPR-Cas9 screen in mice identified genes that regulate interferon-γ (IFNγ) signalling as being key determinants of B16 melanoma sensitivity to combinations of tumour vaccines and anti-PD-L1 immunotherapy [26]. The deletion of *Ptpn2* sensitised B16 tumours to immunotherapy, whereas PTPN2 overexpression rendered cells resistant. The enhanced efficacy of immunotherapy following the loss of PTPN2 was associated with increased granzyme B^+^ cytotoxic CD8^+^ T cell and γδ T cell recruitment to tumours. Mechanistically, increased IFNγ-dependent JAK-STAT signalling led to enhanced antigen processing and presentation in *Ptpn2*^−/−^ B16 cells with subsequent effects on T cell activation [26].

### 2.3. PTPN1 and PTPN2 as Targets in T Cell Cancer Immunotherapies

As described above, PTPN1 and PTPN2 regulate cancer development and responses to immunotherapy. Increasing evidence also points towards an important cell-intrinsic function for these phosphatases in regulating T cell responses to cancer. Wiede and colleagues demonstrated that T cell-specific deletion of *Ptpn2* in mice prevented the formation of tumours induced by p53 loss of heterozygosity [27]. Using OT-I TCR transgenic T cells and an ovalbumin (OVA)-expressing AT3 mammary tumour cell model, these investigators determined that *Ptpn2*^−/−^ CD8^+^ T cell ACT enhanced tumour clearance compared to the control T cell ACT [27]. Similarly, deletion of PTPN2 enhanced the efficacy of HER2-specific mouse CAR-T cells, resulting in superior clearance of HER2-expressing E0771 mammary tumours in vivo and prolonged survival. Enhanced efficacy of PTPN2-deficient CAR-T cells was associated with elevated Lck and STAT5-dependent cytotoxic T lymphocyte (CTL) function and superior STAT5-dependent and CXC chemokine receptor 3 (CXCR3)-mediated homing of CAR-T cells to tumours [27]. The same research team reported a similar T cell-intrinsic role for PTPN1 in regulating tumour immunity. Thus, they determined that PTPN1 expression was elevated in intratumoural T cells compared to splenic counterparts, suggesting that PTPN1 may function as an intracellular immune checkpoint [28,29]. Like PTPN2, PTPN1 regulates STAT5-dependent signalling in T cells, whilst the deletion of PTPN1 enhances the efficacy of CD8^+^ conventional T cell and CAR-T cell ACT in mouse models of cancer [29]. In proof-of-principle studies, the deletion of PTPN1 or PTPN2 also enhanced human CAR-T cell function in vitro [27,29], suggesting that these phosphatases may represent valid targets for future improvements in therapeutic CAR-T design.

### 2.4. Dual PTPN1-PTPN2 Inhibitors as Cancer Therapeutics

The data described above suggest that the use of small molecule inhibitors to block PTPN1/2 function has the potential to exert anti-cancer effects via acting directly on cancer cells and by promoting anti-tumour immunity. Furthermore, acute pharmacological inhibition may reveal distinct effects from the genetic deletion of phosphatases. Thus, gene knockout approaches may result in some degree of functional compensation by other phosphatases, whilst phenotypes associated with the loss of protein expression may be independent of phosphatase catalytic activity. Consistent with the role of the phosphatases in limiting T cell anti-cancer responses, as described using gene knockout models, PTPN1 and PTPN2 inhibitors have been used to enhance anti-tumour immunity and sensitise tumours to other immunotherapy modalities in pre-clinical models [27,29,30]. In recent studies, dual inhibitors that target both phosphatases have been shown to have potent anti-tumour effects in mouse models [31,32]. A small molecule inhibitor, ABBV-CLS-484 [31], and a related compound-182 [32] inhibit PTPN1/PTPN2 with high selectivity over other phosphatases and mediate anti-tumour effects via direct effects on cancer cells and via the enhancement of NK and T cell recruitment and effector function within tumours (Table 1). ABBV-CLS-484 was shown to have broad effects on a range of immune cells, including T cells, natural killer (NK) cells, macrophages and dendritic cells, and had efficacy in experimental settings in which T cell immunity was insufficient, e.g., in MHC1-deficient tumour models [31]. Furthermore, both compounds show efficacy in mouse models that are resistant to PD-1 blockade, without provoking overt inflammation or symptoms of autoimmunity. These data suggest that dual blockade of PTPN1/PTPN2 might represent an approach to overcome tumour immune evasion mechanisms. A phase I trial of the use of ABBV-CLS-484 as a monotherapy, or in combination with anti-PD-1 or tyrosine kinase inhibitors (TKIs), in cancer patients with locally advanced or metastatic solid tumours, is currently underway (NCT04777994).

## 3. PTPN6 and PTPN11

### 3.1. Roles of PTPN6 and PTPN11 in Cell Signalling

PTPN6 and PTPN11 are related Src Homology 2 (SH2) domain-containing PTPs (Figure 1) frequently termed SHP-1 and SHP-2, respectively (reviewed in [33,34]). PTPN6/SHP-1 is predominantly expressed within the haematopoietic system, whereas PTPN11/SHP-2 is ubiquitously expressed. The role of PTPN6 in the regulation of the immune response has been widely studied through the analysis of the spontaneously arising *motheaten* and *motheaten viable* mouse strains, which lack PTPN6 expression or have reduced PTPN6 activity, and, more recently, through mouse strains with lineage-specific deletion of *Ptpn6*. In the complete absence of PTPN6, mice succumb to a severe autoinflammatory disease that is driven by the combined effects of inflammatory neutrophils, macrophages, dendritic cells, B and T cells [35,36,37]. In T cells, PTPN6 is a negative regulator of Lck-dependent TCR signals [38] and interleukin (IL)-4-driven STAT6-dependent signalling [36]. PTPN11 is essential for mouse embryonic development [39] and subsequent lymphopoiesis [40]. In contrast to other phosphatases, PTPN11 has a predominantly positive regulatory role in cell signalling, acting to enhance RAS/MAPK activation. Activating mutations in *PTPN11* underlie ~50% of the cases of Noonan Syndrome, a RASopathy that presents with skeletal malformations and congenital heart disease [41,42].

### 3.2. PTPN11 Is an Oncogenic Phosphatase

Activating mutations in *PTPN11* have been associated with the development of leukaemia through their effects on RAS activation [43,44,45,46]. Furthermore, cancers driven by mutant KRAS are dependent on PTPN11 expression [47,48,49]. Subsequently, the anti-cancer effects of allosteric PTPN11 inhibitors have been widely assessed in pre-clinical models and are under evaluation in early-stage clinical trials for the treatment of cancer (reviewed in [50,51]). As of December 2023, at least ten PTPN11 inhibitors have reached Phase I/II clinical trials for the treatment of solid tumours either as monotherapies or in combination with other anti-cancer drugs, including TKIs and immune checkpoint inhibitors (Table 1). Encouragingly, the results from a Phase 1 trial indicated that the use of the allosteric PTPN11 inhibitor PF-07284892 was able to overcome resistance to diverse TKIs in a range of cancer types. Specifically, therapeutic effects of combined PF-07284892 + TKIs were seen in individuals with *EML4-ALK* fusion-positive lung cancer, BRAF^V600E^-mutant colorectal cancer, *KRAS^G12D^*-mutant ovarian carcinoma and *GOPC-ROS1* fusion-positive pancreatic cancer [49].

An important question is whether PTPN11 inhibitors exert their anti-tumour function solely via effects on cancer cells or whether they also influence the immune response to tumours. Studies demonstrated that an allosteric PTPN11 inhibitor, RMC-4550, reduced CT26 tumour growth in immunocompetent mice but not in recombinase activating gene 2 (RAG2)-deficient mice, indicating a requirement for adaptive lymphocytes in mediating the protective effect [52]. In this setting, PTPN11 inhibition induced a shift in tumour microenvironment (TME) myeloid cell populations towards an inflammatory, anti-tumour phenotype, in part via effects on colony-stimulating factor 1 (CSF1) signalling pathways [52]. Similarly, the treatment of mice with an alternative allosteric PTPN11 inhibitor, SHP099, resulted in enhanced T cell recruitment to orthotopic non-small cell lung carcinoma (NSCLC) nodules [53]. Antibody-mediated depletion of either CD4^+^ or CD8^+^ T cells diminished the anti-tumour effects of SHP099, suggesting an important role for T cell responses in the protective effects of PTPN11 inhibition. Of note, the growth of B16-F10 melanomas was suppressed in mice with specific deletion of *Ptpn11* in myeloid lineages compared to control strains [54]. Improved control of tumour growth in mice with myeloid-specific *Ptpn11*-deficiency was associated with decreased myeloid-derived suppressor cell activity and enhanced activation of tumour-infiltrating T cells [54]. In contrast, T cell-specific deletion of *Ptpn11* does not improve T cell anti-tumour responses [55], indicating that the effects of PTPN11 inhibitors on T cell responses in cancer settings are likely to be indirect and secondary to the modulation of myeloid cell phenotypes.

### 3.3. PTPN6 Acts as Tumour Suppressor

PTPN6 predominantly serves as a negative regulator of signalling. Hypermethylation of *PTPN6* is frequently seen in acute lymphoblastic leukaemia [56], multiple myeloma [57] and mantle cell and follicular lymphomas [58], resulting in the loss of PTPN6 protein expression. Reduced PTPN6 function is associated with enhanced activatory kinase signalling in haematological malignancies. More recently, a similar role for PTPN6 as a tumour-suppressor in solid tumours has also been postulated (reviewed in [59]). Indeed, loss of PTPN6 is associated with poor prognosis in hepatocellular carcinoma (HCC), whilst in healthy hepatocytes and HCC cell lines, PTPN6 inhibits the JAK-STAT, nuclear factor (NF)-κB and Akt-dependent signalling pathways [60]. PTPN6 has been reported to be a negative regulator of epithelial–mesenchymal transition (EMT) and metastasis in HCC [61]. In this regard, PTPN6 suppresses transforming growth factor beta (TGFβ)-dependent STAT5 phosphorylation and subsequent EMT features in HCC cell lines [61].

### 3.4. Role of PTPN6 and PTPN11 in PD-1 Signalling and Function

A dominant role for PD-1-mediated suppression of immune responses is a feature of many cancers. Receptor ligation leads to the recruitment of PTPN6 and PTPN11 to an immunoreceptor tyrosine-based switch motif (ITSM) in the PD-1 intracellular tail [62]. Thus, the recruitment of phosphatases and the subsequent dephosphorylation of CD28 has been suggested to be important in mediating the inhibitory effects of PD-1 signalling in T cells [63]. Furthermore, disruption of the ITSM region prevented PD-1-mediated inhibition of TCR-driven proliferation and cytokine production [62], whilst a vital role for PTPN11 in immune checkpoint receptor function in tumour-infiltrating leukocytes (TILs) has been suggested [64,65,66]. However, T cell-specific deletion of *Ptpn11* does not impact T cell responses to PD-1 [55], whereas the concomitant deletion of both *Ptpn6* and *Ptpn11* in T cells has been reported to abrogate the anti-tumour efficacy of PD-1 blockade in the MC38 colorectal cancer model [67] but not impede PD-1 function in other settings [68]. These data suggest functional redundancy for PTPN6 and PTPN11 in PD-1 function.

### 3.5. Targeting PTPN6 to Enhance T Cell Immune Responses to Cancer

A number of studies have addressed the question of whether and how PTPN6 influences T cell responses in cancer. Conditional deletion of *Ptpn6* renders conventional T cells resistant to the inhibitory effects of regulatory T cells (Tregs) [69], a property that may be beneficial in the suppressive TME. Knockdown of *Ptpn6* expression using short hairpin RNA (shRNA) enhanced OT-I TCR transgenic CD8^+^ T cell responses to B16 melanomas expressing OVA variant proteins as tumour-associated antigens [70]. In particular, the recruitment of *Ptpn6*-deficient OT-I T cells to tumours expressing low-affinity antigen was enhanced compared to control cells. Furthermore, combining *Ptpn6*^−/−^ OT-I T cell ACT with anti-PD-1 treatment resulted in superior control of tumours expressing low-affinity TAA compared to control ACT + PD-1 combinations [70]. Previous studies demonstrated an enhanced capacity for *motheaten* F5 TCR transgenic T cells compared to control F5 T cells to control the growth of B16-NP68 tumours in ACT experiments [71], further adding to the evidence for a negative regulatory function of PTPN6 in the control of T cell anti-cancer responses.

The role of PTPN6 in CAR-T cells has also been assessed. The deletion of *PTPN6*, using CRISPR-Cas9 technology, has been reported to enhance the cytolytic capacity of CD133-targeting CAR-T cells in vitro and anti-tumour activity in vivo [72]. Of note, enhanced PTPN6 expression has been associated with functional exhaustion of CAR-T cells [73]. However, rather than being a driver of exhaustion, an alternative possibility is that PTPN6 recruitment to CAR intracellular domains may act to balance T cell activation and prevent terminal exhaustion [74]. A further factor to consider is that PTPN6 recruitment to CAR cytosolic signalling domains appears to selectively impede T cell inflammatory cytokine production, and thereby reduce the incidence of cytokine release syndrome without impeding anti-tumour activity [75]. Therefore, further studies will be required to determine whether deleting PTPN6 has a net beneficial effect on the outcome of CAR T cell therapies. 

## 4. PTPN22

### 4.1. Roles of PTPN22 in T Cell Signalling

PTPN22 is a member of the proline-, glutamic acid-, serine- and threonine-enriched (PEST) group of phosphatases, which also includes PTP-PEST (PTPN12) and PTPN18. Alternative names for PTPN22 include PEST-domain enriched phosphatase (PEP) and Lymphoid phosphatase (LYP). PTPN22 is predominantly expressed in the cytoplasm of cells of haematopoietic origin. Human and mouse PTPN22 proteins share ~70% amino acid sequence identity, with the N-terminal catalytic domain being the most highly conserved region. Following the catalytic domain and a long “interdomain” region, in the C-terminus of PTPN22, there are four proline-enriched domains termed P1-P4 that regulate protein interactions and turnover (Figure 1). Of note, the P1 region regulates the association of PTPN22 with the Src Homology 3 (SH3) domain of C-terminal Src kinase (CSK) [76], whilst single nucleotide polymorphisms (SNPs) in this region, which disrupt PTPN22-CSK binding, have been identified as genetic risk factors for the development of autoimmune diseases, such as rheumatoid arthritis, type I diabetes and systemic lupus erythematosus [77,78,79].

In T cells, PTPN22 serves as a negative regulator of antigen receptor signalling and influences lymphocyte function-associated antigen 1 (LFA-1)-dependent adhesion (reviewed in [80,81,82]). In brief, PTPN22 dephosphorylates activatory tyrosine residues in kinases, such as Lck and zeta chain-associated protein kinase 70 (ZAP70), thereby dampening proximal TCR signals, as summarized in Figure 2. In the absence of PTPN22, the threshold for activation of T cells following TCR triggering is reduced [83,84], particularly in response to low-affinity TCR ligands [85]. Subsequently, both mouse and human PTPN22-deficient T cells respond more robustly to weak antigenic stimulation than PTPN22-sufficient counterparts [85,86,87]. Furthermore, enhanced TCR-induced IL-2 secretion renders PTPN22-deficient T cells less susceptible to the anti-proliferative effects of the immunosuppressive cytokine TGFβ than wild-type cells [88].

### 4.2. Deletion of PTPN22 Improves T Cell Responses to Cancer

These data suggest a mechanistic basis for the regulation of autoimmunity by PTPN22 but also imply that PTPN22 expression or activity can be manipulated to enhance T cell reactivity in cancer. In this regard, the growth of transplanted MC38 colon carcinoma tumours is suppressed in PTPN22-deficient mice compared to control animals, particularly in the context of PD-1 immune checkpoint blockade [89,90]. Improved control of tumour growth in *Ptpn22*^−/−^ mice is associated with enhanced cytotoxic T cell infiltration into tumours, increased inflammatory cytokine production and alterations in myeloid cell populations. Similar results were reported for knock-in mice that express the autoimmune disease-associated PTPN22 R619W (equivalent to human R620W) variant [89,91]. Furthermore, evidence suggests that expression of the PTPN22 R620W variant is associated with enhanced responses to immune checkpoint therapies in human cancer patients [89,90]. Together, these data provide evidence that PTPN22 is a key negative regulator of anti-tumour immunity in both mice and humans.

Several approaches have been used to address the question of whether PTPN22 can be targeted to enhance cancer immunotherapy. The deletion of PTPN22 did not improve the in vivo anti-tumour activity of murine HER2-specific CAR-T cells [92], reflecting a lack of effect of PTPN22-deficiency on very high-affinity T cell responses [85,86]. Nonetheless, adoptive T cell transfer studies, using TCR transgenic T cells and ID8 ovarian carcinoma and EL4 lymphoma cells modified to express model antigens, demonstrated an enhanced capacity of PTPN22-deficient CD8^+^ T cells to clear established tumours in mice [88,93]. The enhanced anti-tumour function of PTPN22-deficient CD8^+^ T cells was particularly evident in response to tumours expressing low-affinity TCR antigens and was associated with elevated cytokine secretion and direct TCR-dependent cytotoxicity. Importantly, memory-phenotype PTPN22-deficient CD8^+^ T cells, polarized in vitro in the presence of IL-15, had the capacity to clear tumours and were retained in mice for months after tumours became undetectable [93]. However, recent studies suggest that under conditions of chronic stimulation, PTPN22-deficient effector CD8^+^ T cells can become exhausted and may function less well than wild-type T cells in anti-cancer responses [94]. Therefore, balancing an enhanced capacity for effector responses with an increased propensity for exhaustion is a key concern for future targeting of PTPN22 in adoptive T cell cancer therapies.

### 4.3. PTPN22 Is a Druggable Target

The pharmacological inhibition of PTPN22 may also represent a valid approach to improve anti-cancer immune responses. Ho and colleagues reported that treatment with a PTPN22 selective inhibitor, L-1, reduces the growth of MC38 and CT26 colon carcinomas in wild-type but not PTPN22-deficient mice [90]. L-1 treatment in WT mice phenocopies PTPN22-deficiency in terms of enhancing T cell and myeloid cell-dependent tumour immunity. Importantly, L-1 appears to be non-toxic and in combination with PD-1 blockade, results in improved control of tumour growth [90]. More recently, similar data were reported using chemically distinct PTPN22 inhibitors termed D14 and D34 [95], adding to the evidence for the utility of targeting PTPN22 to improve cancer immunity.

## 5. Roles for Other Tyrosine Phosphatases in Cancer Biology and Tumour Immunity

### 5.1. PTPN3, PTPN4 and PTPN13

PTPN3 AND PTPN4 are related phosphatases that contain band 4.1, ezrin, radixin, moesin (FERM) and PSD-95, Dlg, ZO-1 (PDZ) domains, in addition to a C-terminal PTP domain (Figure 1). Although implicated in the regulation of TCR signal transduction, knockout mouse studies indicate that the deletion of PTPN3 and PTPN4 alone or in combination does not impact T cell responses [96,97,98]. Nonetheless, reports have suggested that shRNA-mediated knockdown of *PTPN3* enhances human T cell responses to ovarian cancer [99] and small-cell lung carcinoma [100] xenografts in mice. PTPN3 may also negatively regulate dendritic cell function in cancer [101] and has tumour-suppressor activity, which is independent of phosphatase activity, by potentiating TGFβ-driven growth inhibitory responses in HCC cell lines [102].

PTPN13, called PTPL1, FAP1 and PTP-BL in some of the literature, has lower sequence homology to PTPN3 and PTPN4 but also contains FERM and PDZ domains (Figure 1). PTPN13 has been implicated in the regulation of PI3K signalling [103] and apoptosis via the regulation of Fas death receptor expression [104]. Roles for PTPN13 in both promoting and suppressing tumour development have been reported (reviewed in [105]). Thus, PTPN13 may suppress oncogenic Src signalling [106], whilst the loss of *PTPN13* expression [107] or germ-line mutations in *PTPN13* [108] have been described in NSCLC and acute lymphoblastic leukaemia, respectively. PTPN13 has been described as a STAT phosphatase and regulates CD4^+^ T helper cell differentiation in mice [109]. However, to date, the role of PTPN13 in the regulation of cancer immunity has yet to be determined.

### 5.2. PTPN12

PTPN12, also called PTP-PEST, is a cytosolic phosphatase that has a tumour-suppressor function in breast cancer [110,111,112] and renal cell carcinoma [113]. PTPN12 function or expression is frequently lost in TNBC, whereas PTPN12 activity inhibits HER2 and epidermal growth factor receptor (EGFR) signalling pathways and transformation. The re-expression of PTPN12 suppresses TNBC growth and metastasis in vivo [112] and restrains RTK-dependent signalling [111]. In T cells, PTPN12 was originally described as a negative regulator of TCR-induced activation [114] but has subsequently been shown to be dispensable for primary T cell responses [115]. Instead, PTPN12 acts as a positive regulator of secondary T cell responses via the suppression of anergy [115]. The role of PTPN12 in the regulation of anti-cancer immune responses has yet to be defined.

### 5.3. CD45

CD45 is a receptor-like PTP (Figure 1) and is one of the most abundant glycoproteins expressed on the surface of haematopoietic cells (reviewed in [116]). CD45 can suppress the development of T cell lymphomas in mice expressing an active Lck transgene (LckF505) [117] and myeloproliferative phenotypes in mice expressing activating mutations in the FLT3 RTK [118]. Furthermore, loss-of-function mutations in *PTPRC*, which encodes CD45, have been detected in acute lymphoblastic leukaemia [119], suggesting that CD45 may have tumour-suppressor activity. CD45 has dual positive and negative regulatory functions in T cells via effects on Lck phosphorylation [120,121,122,123,124]. In particular, CD45 exclusion from the TCR–peptide–MHC interface and segregation from kinases is thought to be important to enable activatory signals in T cells [120]. Similarly, the exclusion of CD45 from the vicinity of CAR-surface antigen pairing is important for enabling signalling in therapeutic CAR T cells [125]. The expression of longer CD45 isoforms, such as CD45RABC, results in enhanced CAR T cell signalling and activation compared to cells expressing smaller CD45RO isoforms [125], suggesting that the manipulation of CD45 isoform expression could be used to tune CAR T cell activity.

## 6. Concluding Remarks

There has long been an understanding that PTPs play an important role in the regulation of tumour cell signalling, the development of cancer and the regulation of immune responses. As described in the current work, a wealth of data has demonstrated that a number of PTPs can be targeted to improve cancer responses via effects on therapeutic T cells and endogenous immune responses as well as direct anti-cancer effects, as summarized in Figure 3. In recent years, there has been an appreciation that PTPN family members, such as PTPN1, PTPN2 and PTPN22, may serve as intracellular immune checkpoints, analogous to the function of cell surface checkpoint receptors. As described in this review, approaches to manipulate PTPN family expression have resulted in enhanced therapeutic function of anti-cancer T cells and ACT responses in pre-clinical tumour models, raising the hope that PTP family members might be targeted to improve therapeutic CAR-T or conventional TCR-expressing T cell therapies in patients. Of note, patients are currently being recruited to a Phase 1 trial assessing the impact of the deletion of the inhibitory signalling protein Cish on TIL ACT therapy for lung cancer (NCT05566223). It is hoped that similar studies to assess the impact of the deletion of inhibitory PTPs in therapeutic T cells will proceed in the near future.

For many years, it was thought that tyrosine phosphatases were poor drug targets due to the similarity in PTP catalytic domains between different family members and the potential for off-target effects. However, the catalytic site-targeting drug ABBV-CLS-484 has a high degree of selectivity, inhibiting PTPN1 and PTPN2 at nanomolar concentrations and only significantly inhibiting PTPN9, but not other PTP family proteins, at millimolar levels [31]. These data suggest that targeting catalytic sites remains a viable approach for the development of selective phosphatase inhibitors. Nonetheless, the use of allosteric inhibitors that target unique regions outside the PTP domain has expanded the potential for selectively targeting these enzymes. Of note, allosteric PTPN11 inhibitors have shown high specificity over the related PTPN6 whilst retaining anti-cancer effects in pre-clinical studies and early-stage clinical trials.

A further key consideration for the development and use of any novel drug is toxicity. Allosteric PTPN11 inhibitors appear to have an acceptable toxicity profile in stage 1 clinical trials, although further dose escalation studies are ongoing [49,126]. Of note, trials assessing the efficacy of several PTPN1 inhibitors for the treatment of type 2 diabetes were halted due to low efficacy and toxicities, including vomiting and diarrhoea [127,128]. Pre-clinical studies suggest low toxicity of novel dual PTPN1/2 inhibitors in mice; it is hoped that similar safety profiles are revealed in ongoing clinical trials [31]. However, given the reported toxicities of PTPN1-selective inhibitors and the key role that PTPN1 plays in metabolic regulation and PTPN2 plays in immune responses, it is likely that these will not be without any side effects. In summary, the use of highly specific phosphatase inhibitors with both anti-cancer and immune-stimulating capacity has shown great potential in mouse models and has now reached early-stage clinical trials. The hope is that these advances in our understanding of PTP biology will progress to the development of improved cancer therapies in the coming years.

## Figures and Tables

**Figure 1 cells-13-00231-f001:**
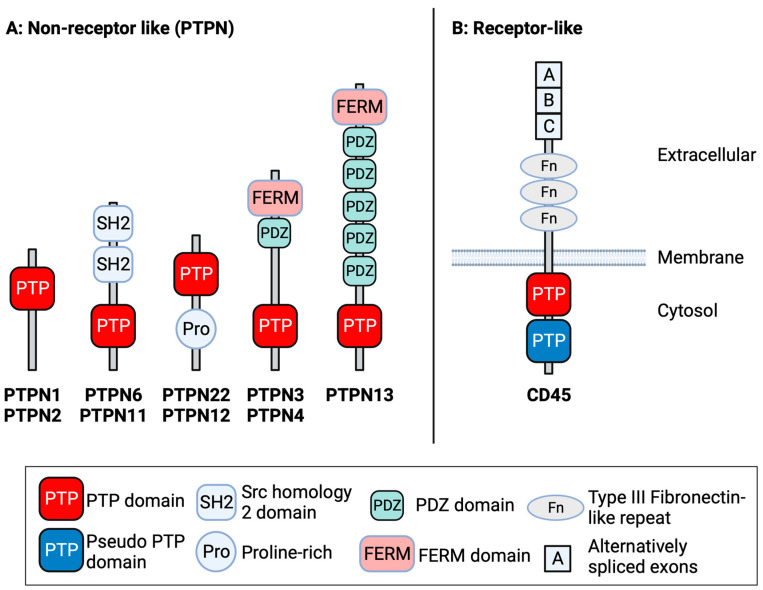
Domain structure of protein tyrosine phosphatases (PTPs). Overview of the basic structure of non-receptor PTPs (PTPNs) and CD45, described in the current review. PTPNs with shared structural domains (e.g., PTPN6 and PTPN11) are depicted as such. Image created with BioRender.com.

**Figure 2 cells-13-00231-f002:**
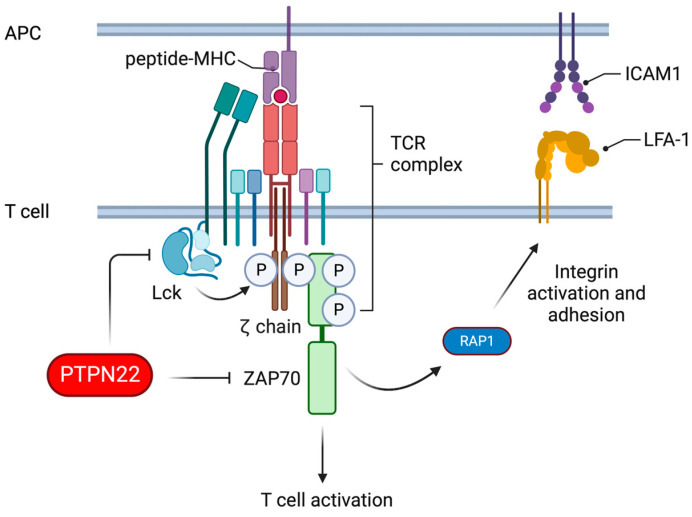
PTPN22 regulates proximal TCR signalling and LFA-1-dependent adhesion. Upon TCR ligation by peptide-MHC complexes, Lck phosphorylates the immunoreceptor tyrosine-based activation motifs of the CD3/ζ-chain complex, resulting in the recruitment and activation of ZAP70. ZAP70 subsequently induces downstream TCR signalling and T cell activatory signals as well as “inside-out” signalling via the GTPase Rap1. Rap1 activation results in enhanced LFA-1-dependent adhesion processes. PTPN22 dephosphorylates tyrosine residues in TCR proximal kinases Lck and ZAP70 to suppress T cell activation. Image created with BioRender.com.

**Figure 3 cells-13-00231-f003:**
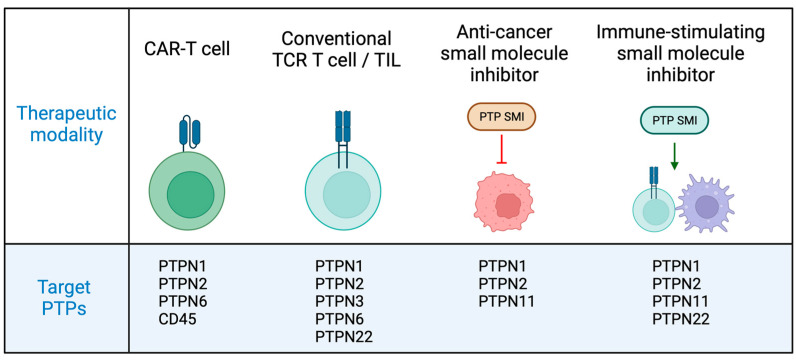
Targeting protein tyrosine phosphatases in cancer therapy. In pre-clinical studies, deletion of several PTPs can improve the functional capacity of therapeutic CAR-T and conventional TCR T cells for ACT approaches. Small molecule PTP inhibitors have shown efficacy in pre-clinical and early-stage clinical trials for the treatment of cancer via direct anti-cancer and immune-stimulatory effects. Image created with BioRender.com.

**Table 1 cells-13-00231-t001:** Development of small molecule PTP inhibitors as anti-cancer drugs. a.s.—active site (competitive) inhibitor. *—Trial discontinued due to lack of interest by sponsor. **—recruiting. ***—trial completed, results not yet reported.

Inhibitor (Mechanism)	Target PTP(s)	Stage of Development	Cancers Targeted
MSI-1436 (allosteric)	PTPN1	Phase I *	Metastatic breast cancer
Compound 8 (a.s.)	PTPN2	Pre-clinical	Mouse models
PTP9 (a.s.)	PTPN2	Pre-clinical	Mouse models
ABBV-CLS-484 (a.s.)	PTPN1/PTPN2	Phase I	Locally advanced/metastatic solid tumours
Compound 182 (a.s.)	PTPN1/PTPN2	Pre-clinical	Mouse models
TNO155 (allosteric)	PTPN11	Phase I/II	Advanced solid tumours
PF-07284892 (allosteric)	PTPN11	Phase I	Advanced solid tumours
RMC-4630 (allosteric)	PTPN11	Phase I **	Metastatic KRAS mutant tumours
BBP-398 (allosteric)	PTPN11	Phase I	Advanced solid tumours with KRAS-G12C
JAB-3068 (allosteric)	PTPN11	Phase I	Advanced solid tumours
JAB-3312 (allosteric)	PTPN11	Phase I/IIa	Advanced solid tumours with KRAS-G12C
RLY-1971 (allosteric)	PTPN11	Phase I ***	Advanced/metastatic solid tumours
HBI-2376 (allosteric)	PTPN11	Phase I **	Advanced solid tumours
ET0038 (allosteric)	PTPN11	Phase I	Advanced solid tumours
ERAS-601 (allosteric)	PTPN11	Phase I/Ib	Advanced solid tumours
BR790 (allosteric)	PTPN11	Phase I/IIa	Advanced solid tumours
L-1 (a.s.)	PTPN22	Pre-clinical	Mouse models
D14/D34 (a.s.)	PTPN22	Pre-clinical	Mouse models

## Data Availability

Not applicable.

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
