# Peer review of "Targeting Protein Tyrosine Phosphatases to Improve Cancer Immunotherapies"

_cells, 2024, doi:10.3390/cells13030231_

Round 1
Reviewer 1 Report
Comments and Suggestions for Authors
In this review the authors summarized information regarding protein tyrosine phosphatase (PTP), their roles, in normal cells, in tumours and their applications as antitumor targets via immunotherapy.
This review collects a large quantity of information, however, it lacks too many things to be really informative, especially in terms of structure of the text and in terms of figures. With such a review, one can expect:
A figure for summarizing role of each PTP in normal tissues/cells, in cancers and in TME. This point is quite unclear here because most of PTPs are expressed in immune cells but also in many normal cells.
An explanation of the selection of PTP presented here ? Why this selection ? Why no comment about all the other PTPs ?
A clear presentation of applications: Which PTP could be really targetable with drugs ? And which could be rather considered for optimizing CAR-T cells ?
A figures for explaining the applications in CAR-T cells and not only with PTPN22, other PTPs are proposed in other chapters.
A table summarizing PTP inhibitors, their target and the development stage (to really appreciate the applicative research on this topic)
An enriched discussion:
There is a very low number of phosphatases in comparison to kinases. Therefore, the risk that drugs targeting PTP could be less selective is high. The question of the risk of toxicity with PTP is still a debate …. But absolutely no word about this question. It is difficult to discuss application without this subject.
Consequently of the previous point, is the application with CAR-T cells more realistic ? And concerning anti-PTP drugs, how is the way for limiting toxicity ? is it by the nature of the inhibition ? is due to the allosteric ? to a transitory effect ?
With all the information presented, what is the hypothesis the authors could suggest in terms of application ? What could be the better PTP to target (their presentation here suggest they are quite redundant and their application could be similar) ?
If the list of information is interesting, nothing is really discussed.
Other remarks:
In the presentation of data, it is important to underscore that experiment with deletion of PTP gene could result in some compensation and that might differ from an inhibition with small molecule. It is important to mention that for really compare information from very different experimental approaches.
2.1 and 3.1 are entitle “role in cell signalling”, the right titles could be “role in normal cells/tissues” because role of PTPs in cancer involves also cell signalling.
Why chapter 5 concerns only tumour immunity ? if so, the first part of the presentation of PTPN12 (lines 343-348) is confusing by presenting role of PTPN12 in cancer cells.
Line 369-370 ?
Author Response
I thank the reviewer for their helpful comments. Below I have addressed their comments (my comments in red).
-------
This review collects a large quantity of information, however, it lacks too many things to be really informative, especially in terms of structure of the text and in terms of figures. With such a review, one can expect … a figure for summarizing role of each PTP in normal tissues/cells, in cancers and in TME. This point is quite unclear here because most of PTPs are expressed in immune cells but also in many normal cells. An explanation of the selection of PTP presented here? Why this selection? Why no comment about all the other PTPs?
- The focus of this review is an overview of ongoing research into PTP family proteins as potential therapeutic targets for the treatment of cancer, with a particular focus on immunotherapies. The review focuses on the functions of 5 PTPN family phosphatases for which a substantial literature in these areas exists, plus a brief overview of the more limited literature on 5 additional PTPs. It is not intended as an exhaustive overview of PTP biology, for which alternative review articles are suggested in the text (Line 39, refs 7-10). The reviewer has requested new figures for each PTP, giving overviews of individual functions in immune cells, the TME and in cancer cells plus additional new figures on anti-cancer applications, CAR T cells and a table. This would result in having a total of 15 figure items (including the current 2 figures) which I believe is not appropriate within a focused review. Nonetheless, I have added a new figure 3 (see below) and a new Table. Furthermore, the roles of the various PTPs in immune cells and cancer cells are described thoroughly in the text.
A clear presentation of applications: Which PTP could be really targetable with drugs? And which could be rather considered for optimizing CAR-T cells? A figures for explaining the applications in CAR-T cells and not only with PTPN22, other PTPs are proposed in other chapters.
- The potential for targeting individual phosphatases via small molecule inhibitor or via deletion in therapeutic T cells is covered extensively in the text. The revised manuscript also contains a new figure 3 to summarize these points.
A table summarizing PTP inhibitors, their target and the development stage (to really appreciate the applicative research on this topic)
- As requested by the reviewer, a new table 1 has been added to summarize the development of PTP inhibitors as cancer drugs.
An enriched discussion:
There is a very low number of phosphatases in comparison to kinases. Therefore, the risk that drugs targeting PTP could be less selective is high. The question of the risk of toxicity with PTP is still a debate …. But absolutely no word about this question. It is difficult to discuss application without this subject. Consequently of the previous point, is the application with CAR-T cells more realistic? And concerning anti-PTP drugs, how is the way for limiting toxicity? is it by the nature of the inhibition? is due to the allosteric? to a transitory effect?
- These important points, concerning inhibitor specificity and the potential for toxicity, are discussed in new text in Section 6 (Lines 402-412 and 420-428 in the revised manuscript).
With all the information presented, what is the hypothesis the authors could suggest in terms of application? What could be the better PTP to target (their presentation here suggest they are quite redundant and their application could be similar)?
- The purpose of the current review is not to propose hypotheses about which PTP represents a “better” target, rather reflect the current state-of-play as objectively as possible. As described in the text and illustrated in the new Table 1, the use of PTPN11 inhibitors is clearly more advanced in terms of clinical development with numerous distinct compounds being tested in Phase 1/2 trials. Targeting PTPs for ACT approaches has yet to reach clinical trial.
Other remarks:
In the presentation of data, it is important to underscore that experiment with deletion of PTP gene could result in some compensation and that might differ from an inhibition with small molecule. It is important to mention that for really compare information from very different experimental approaches.
- These points, concerning the potential for discrepancies between inhibitor studies and gene KO studies, are now summarized in new text (lines 122-127). The effects of inhibitors in mouse studies in the context of the respective knockout mouse studies are described in the text: eg PTPN1/2 – lines 128-130. PTPN22 – lines 318-320. The potential for compensation/redundancy between phosphatases is described where relevant e.g. in the case of the roles for PTPN6 and PTPN11 in PD-1 signalling.
2.1 and 3.1 are entitle “role in cell signalling”, the right titles could be “role in normal cells/tissues” because role of PTPs in cancer involves also cell signalling.
- I agree that PTPs play a key role in cancer cell signalling and have discussed these roles in detail throughout the text. Nonetheless, I prefer to retain the current titles for these sections.
Why chapter 5 concerns only tumour immunity ? if so, the first part of the presentation of PTPN12 (lines 343-348) is confusing by presenting role of PTPN12 in cancer cells.
- The title for section 5 has been changed in line with the reviewer’s query to “Roles for other tyrosine phosphatases in cancer biology and tumour immunity” to better reflect the content.
Line 369-370?
- These lines were left over from the original Cells manuscript template and have been deleted in the revised manuscript.
Reviewer 2 Report
Comments and Suggestions for Authors The manuscript entitled “Targeting Protein Tyrosine Phosphatases to improve cancer immunotherapies” by Salmond is an excellent review of the studies investigating the impact of targeting protein tyrosine phosphates across a multitude of different cancer models. The author did a great job of clearly explaining the, sometimes highly complex, cellular interactions of PTPs and how altering their function impacts the efficacy of different cancer immunotherapies. The author covers a wide breadth of the field in the discussion and the papers referenced are highly relevant. The figures aid nicely to form (figure 1) and function (figure 2) of the discussion and although the material of the review can sometimes be a bit dense, it is simply due to the complexity of the studies being described.
In line 369-370, the author has a note that section 5 (or perhaps more specifically CD45) may not be necessary, but I believe that it nicely and succinctly covers what is known in other relevant PTPs in tumor immunity and is indeed worth keeping in the review.
Great job!
Author Response
Thank you to the reviewer for the positive feedback!
Reviewer 3 Report
Comments and Suggestions for Authors
Recent strides in immunotherapy have revolutionized cancer treatment, offering substantial benefits to numerous patients. However, the efficacy of current immunotherapeutic approaches is limited in certain cancer types, necessitating the identification of additional targets to broaden the spectrum of beneficiaries. Protein tyrosine phosphatases (PTPs) have long been recognized for their pivotal roles in regulating cancer cell biology and immune responses. This review consolidates evidence supporting both the pro-tumorigenic and tumor-suppressor functions of non-receptor PTPs in cancer cells. Additionally, we delve into recent findings indicating that several of these enzymes serve as intracellular immune checkpoints, dampening effective anti-tumor immunity. Notably, we emphasize emerging data supporting the notion that deleting inhibitory PTPs presents a rational strategy to enhance the outcomes of adoptive T cell-based cancer immunotherapies. Furthermore, the authors spotlight recent advancements in the development of PTP inhibitors as potential anti-cancer drugs, underscoring the promising progress in leveraging these molecules for therapeutic interventions in cancer.
Main concerns;
1. A summary chart could enhance the visual representation of PTP inhibitors as potential anti-cancer drugs in your review.
2. While the PTP inhibitors exhibit promising anti-cancer effects, it is imperative to address potential side effects that may accompany their use. The modulation of PTPs can impact various cellular functions, raising concerns about unintended consequences.
Author Response
I thank the reviewer for their helpful comments. My replies to specific comments are below (my comments in red).
--------------
1. A summary chart could enhance the visual representation of PTP inhibitors as potential anti-cancer drugs in your review.
As requested, a new table 1 has now been included summarizing the stages of development of various PTP inhibitors for cancer therapy.
2. While the PTP inhibitors exhibit promising anti-cancer effects, it is imperative to address potential side effects that may accompany their use. The modulation of PTPs can impact various cellular functions, raising concerns about unintended consequences.
I agree this is an important consideration. In line with the reviewers' suggestions, the conclusion section has now been extended with new text to include greater discussion of the importance of inhibitor specificity and the potential for toxicities associated with their use (Lines 402-428 in revised manuscript).
Round 2
Reviewer 3 Report
Comments and Suggestions for Authors
The authors well addressed my previous concerns.